# Immediate Hypersensitivity Reactions to Paraformaldehyde Used as a Dental Material

**Edyta Jura-Szołtys \*, Lesia Rozłucka , Radosław Gawlik and Joanna Glück**

Department of Internal Medicine, Allergology and Clinical Immunology, Faculty of Medical Sciences in Zabrze, Medical University of Silesia, 40-752 Katowice, Poland
\* Correspondence: edytajura@interia.pl; Tel.: +48-32-789-4641

**Abstract:** According to actual recommendations, the use of paraformaldehyde as a dental material should be significantly limited; however, it is still used in certain cases. Its use can cause delayed anaphylaxis, which can be life-threatening. We analyzed 157 patients admitted to an allergological clinic between 2017 and 2022 because of a hypersensitivity reaction after dental treatment. Paraformaldehyde was used in 24 of them. Positive specific IgE levels against paraformaldehyde were found in 12 patients, which constituted 50% of the whole group of patients who were treated with paraformaldehyde. Twelve patients had negative results of specific IgE against paraformaldehyde estimation (the PF group). Between the study and control groups, the anaphylactic reaction after paraformaldehyde application was analyzed from many aspects; the level of certainty of anaphylaxis according to Brighton criteria was significantly higher in the PF-positive group than in controls. None of the patients treated with paraformaldehyde as a dental material was informed by a dentist about this risk and symptoms of anaphylaxis. Patients who received paraformaldehyde during dental treatment should be informed of the possibility, symptoms, and treatment of an anaphylactic reaction, which might occur even 24 h after treatment.

**Keywords:** allergy; anaphylaxis; delayed hypersensitivity; dental material; paraformaldehyde

## 1. Introduction

Allergies are a significant clinical problem in everyday medical practice, including dentistry. The most common allergic reactions in a dentist's office include types I (immediate) and IV (delayed) reactions according to the Gell and Coombs classification [1]. Among the allergic reactions, the most significant, life-threatening reaction is anaphylaxis, which belongs to type I hypersensitivity. It requires immediate intervention because, if left untreated, it may lead to the patient's death. In 2021, The European Academy of Allergy and Clinical Immunology Anaphylaxis Multidisciplinary Task Force updated the 2014 guidelines for the diagnosis and treatment of anaphylaxis [2]. Symptoms of anaphylaxis may affect different systems and organs. From the data available in the literature, it appears that the dominant symptoms of anaphylaxis include the skin and mucous membranes (>90%) and are followed by respiratory or cardiovascular symptoms (>50%) [3]. Sometimes, anaphylaxis occurs without skin and mucosal symptoms, making an accurate diagnosis difficult. In those cases, despite the obvious symptoms of anaphylaxis, the causative agent is not established [4]. The Sampson criteria from 2006 have been proposed in order to facilitate the diagnosis of anaphylaxis [5]. The certainty of anaphylaxis may be assessed by Brighton Collaboration criteria [6].

In case of clear symptoms of anaphylaxis, it is relatively easy to associate a reaction with the exposure to a specific allergen. On the other hand, in delayed hypersensitivity reactions of type IV, such as contact stomatitis, the course of the disease is chronic, which makes it difficult to identify the causative allergen. The symptoms of type IV reaction include burning sensation in the mouth, pain, paresthesia, numbness, itching, unpleasant

taste, and hypersalivation. Physical examination reveals swelling, redness, sensitivity to pressure, abrasions, ulcerations, changes resembling leukoplakia or lichen planus, and a geographic tongue [7].

Allergic reactions in a dentist's office may appear after administering local anesthetics and nonsteroidal anti-inflammatory drugs; after contact with metals (nickel, palladium, gold, silver, cobalt, mercury, titanium, and platinum), acrylic resins, latex, materials used in canal filling, eugenol-containing agents, impression materials, and resins used in composite materials; as well as after the use of dental materials with paraformaldehyde [8]. Paraformaldehyde is a compound widely used in the cosmetics industry, as a disinfectant and preservative, and in dentistry, as a component of dental material used for devitalizing the dental pulp, paste used in endodontic practice, and other materials such as epoxy resin cements, which do not contain formaldehyde, but they may release minimal levels of formaldehyde during the setting reaction [9]. Hypersensitivity reactions to paraformaldehyde most frequently occur as contact dermatitis [10]. However, in the literature, there are reports of an immediate IgE-mediated reaction after the use of dental materials containing paraformaldehyde [11,12]. IgE-mediated allergic reactions to paraformaldehyde, including anaphylaxis, are specific. They may not immediately appear after contact with the allergen but often occur several hours after contact with paraformaldehyde. This is due to the fact that the dental materials used for devitalizing the dental pulp containing paraformaldehyde are gradually and slowly absorbed from the injection site. Often, materials used for devitalizing the tooth pulp and local anesthetics are administered to the patient during one visit. In such patients with allergic reaction only to paraformaldehyde, the patient is unreasonably forbidden to use local anesthetics, so it is important to perform a differential diagnosis. It is worth noticing that a great number of adverse reactions following the use of local anesthetics are due to the toxic effects of these drugs, administration of too high a dose, or accidental intravenous administration. In addition, psychogenic reactions or vasovagal syncope is often misinterpreted as an allergic reaction [13].

In the diagnosis of immediate hypersensitivity reactions to formaldehyde, due to the lack of standardization and the possibility of causing systemic symptoms, prick and intradermal paraformaldehyde tests are not performed. The most useful method is the determination of allergen-specific IgE antibodies for formaldehyde. The diagnosis of delayed hypersensitivity is based on patch tests with 1% formaldehyde solution.

The aim of the study was to assess the frequency of immediate-type hypersensitivity and analyze the course of the reaction after the use of dental materials containing paraformaldehyde.

## 2. Materials and Methods

This single-center retrospective observational case-control study included patients referred to the Department of Internal Medicine, Allergology and Clinical Immunology at Medical University of Silesia in Katowice between 2017 and 2022 because of hypersensitivity reaction in dentistry with suspicion of local anesthetics hypersensitivity. Among all these patients, only those who received paraformaldehyde were further investigated. The patients with specific IgE against paraformaldehyde constituted the case group, and the patients without specific IgE against paraformaldehyde constituted the PF group. In both groups, the following data were obtained from the medical records: sex, age, diagnosis of atopic diseases such as seasonal and/or persistent allergic rhinitis, asthma, and hypersensitivity to other drugs. Moreover, the data on symptoms that developed in dentistry (urticaria, angioedema, pruritus, general erythema, dyspnea, sensation of throat, fainting, fall in blood pressure, loss of consciousness, and gastrointestinal symptoms), time of appearance, duration, and treatment were carefully collected. The Brighton Collaboration criteria of certainty of anaphylaxis and Sampson criteria were also calculated [5,6].

Serum samples were routinely collected in all patients admitted to the department because of hypersensitivity reaction in dentistry. Two milliliters of blood was drawn from a peripheral vein to a sample tube without anticoagulant, and then the serum was cen-

trifuged (3000 rpm for 10 min at 4 °C) before further analysis. Quantitative assessment of serum-specific IgE assays against formaldehyde was performed using ImmunoCAP (Thermo Fisher Scientific). The results of sIgE were divided into 7 levels: 0 (<0.35 IU/mL), 1 (0.35–0.70 IU/mL), 2 (0.70–3.50 IU/mL), 3 (3.5–17.5 IU/mL), 4 (17.5–50 IU/mL), 5 (50–100 IU/mL), and 6 (>100 IU/mL). An sIgE level of <0.35 IU/mL was considered to be negative.

Some patients underwent patch tests with 1% formaldehyde. Patch testing was performed by the use of Finn Chambers of 8 mm. Formaldehyde 1% solution (0.30 mg/cm$^2$) was provided by Chemotechnique MB Diagnostics F-002C and applied with a micropipette (15 μL) to a filter paper in Finn Chambers. The negative control was conducted with 15 μL 0.9% NaCl solution. The patch tests were applied to the upper back for 48 h. Two readings were performed: after 48 and 72 h. The results were scored as +, ++, +++, or negative. The +, ++, and +++ results were classified as positive.

The study was not submitted for approval to the The Bioethics Committee because it was based only on the retrospective analysis of medical records of patients hospitalized in the clinic.

*Statistical Analysis*

Results are expressed as absolute numbers and percentages for frequencies and median with interquartile ranges. The nonparametric Mann–Whitney U rank sum and Fisher exact tests were used. All analyses were performed with a software package (STATISTICA 13.3, StatSoft Poland, Kraków, Poland). *p* values less than 0.05 were considered significant.

## 3. Results

Between 2017 and 2022, 157 patients with suspicion of local anesthetics allergy were allergologically assessed. Among them, 24 subjects experienced hypersensitivity reaction after stomatological treatment using a preparation containing paraformaldehyde and local anesthetics. A total of 12 patients (3 men, 9 women, PF+ group) had positive specific IgE against paraformaldehyde, i.e., above 0.35 kU/L, and 12 patients (1 man, 11 women, PF–group) had negative results of specific IgE against paraformaldehyde estimation, i.e., below 0.35 kU/L (class 0) (Figure 1).

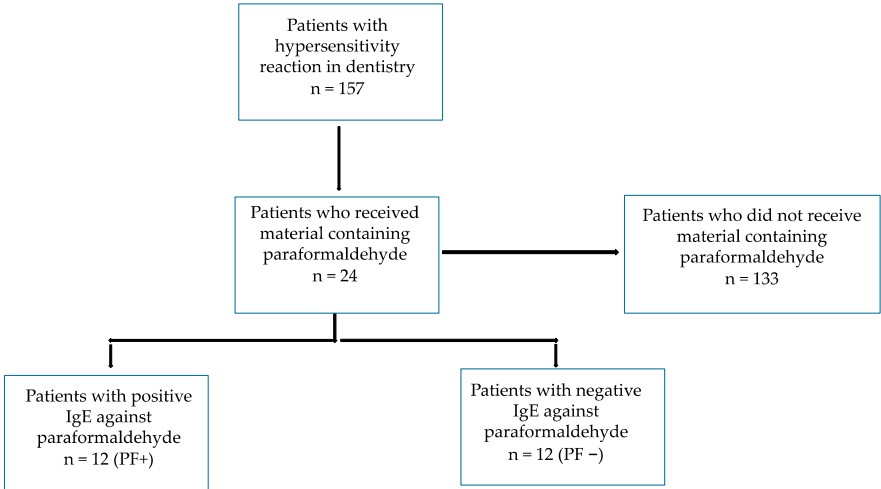

**Figure 1.** Flow chart of recruited cases.

The specific IgE against paraformaldehyde ranged between 0.5 and 100 kU/L, median was 2.31, and interquartile range was 0.9–2.35 kU/L. Two (16.7%) results were within class 1, four (33.3%) within class 2, three within class 3 (25%), two (16.7%) within class 4, and one (8.3%) within class 6 (Figure 2).

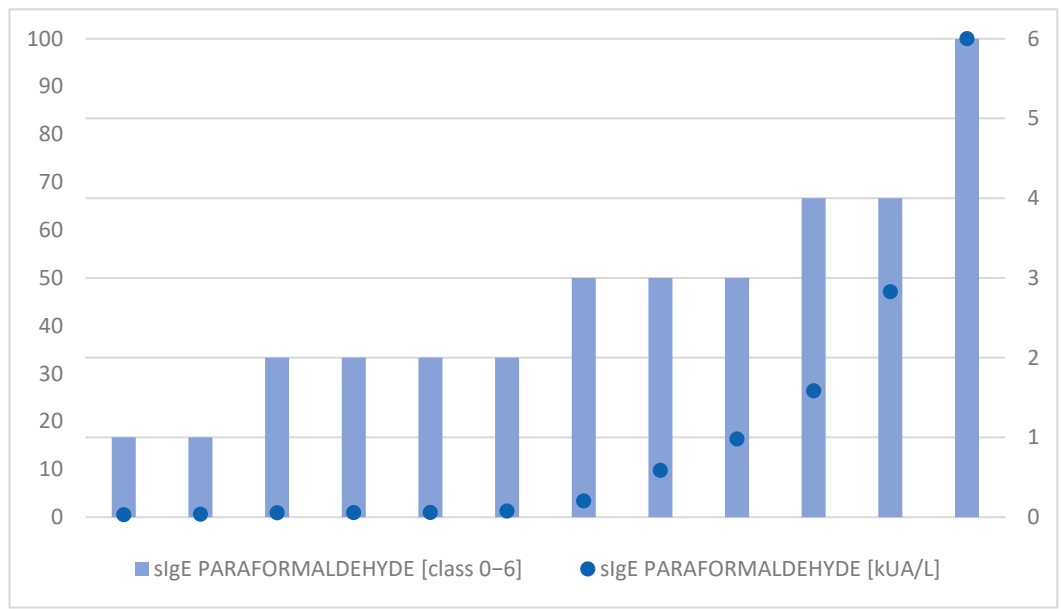

**Figure 2.** Level of specific IgE against paraformaldehyde.

*Description of PF-Positive Group*

Patients from the PF-positive group did not differ from controls with regard to sex, age, atopic comorbidities, asthma, or history of allergy to other drugs. In the PF positive group, general pruritus and erythema were significantly more common than in the PF negative group. The levels of certainty of anaphylaxis according to Brighton criteria were significantly higher in the PF-positive group than in controls ($p = 0.03$). Two patients within the PF+ group had also positive patch tests with paraformaldehyde. One patient within the PF+ group had two different reactions to paraformaldehyde—type IV reaction presented as contact dermatitis with exanthema and type I immediate reaction including angioedema and dyspnea. Patients' clinical characteristics are presented in Table 1.

**Table 1.** Patients' clinical characteristics.

| Group | PF+ ($n = 12$) | PF− ($n = 12$) | $p$ Value |
|---|---|---|---|
| Men/Women | 3 (25%) | 1 (8.3%) | ns |
| Age, years | 40 (22–54) | 44.5 (19–62) | ns |
| Atopy | 3 (25%) | 2 (16.7%) | ns |
| Asthma | 2 (16.7%) | 2 (16.7%) | ns |
| SAR | 3 (25%) | 2 (16.7%) | ns |
| PAR | 1 (8.3%) | 0 | |
| Drug allergy in history | 3 (25%; NSAIDs, ampicillin, statins) | 3 (25%; penicillin, 2 × NSAIDs) | ns |
| Time to reaction | | | |
| <1 h | 2 (16.7%) | 6 (50%) | ns |
| 1–24 h | 7 (58.3%) | 4 (33.3%) | ns |
| >24 h | 2 (16.7%) | 1 (8.3%) | ns |
| >1 h | 9 (75%) | 5 (41.7%) | $p = 0.05$ |
| Urticaria | 8 (66.7%) | 6 (50%) | ns |
| AE | 7 (58.3%) | 6 (50%) | ns |
| Erythema | 10 (83.3%) | 3 (25%) | $p = 0.006$ |
| Pruritus | 12 (100%) | 8 (66.7%) | $p = 0.047$ |

**Table 1.** *Cont.*

| Group | PF+ (*n* = 12) | PF− (*n* = 12) | *p* Value |
|---|---|---|---|
| Dyspnea | 5 (41.7%) | 2 (16.7%) | ns |
| Feeling of the obstruction of the throat | 7 (53%) | 3 (25%) | ns |
| Weakness | 3 (25%) | 4 (33.3%) | ns |
| Blood pressure decrease | 1 (8.3%) | 3 (25%) | ns |
| Loss of consciousness | 1 (8.3%) | 2 (16.7%) | ns |
| Immediate reaction | 11 (92%) | 10 (83%) | ns |
| Gastric symptoms | 0 | 1 (8.3%) | ns |
| Brighton level (*n*, %) | Level 1–5 (41.6%) Level 2–6 (50%) Level 5–1 (8.3%) | Level 2–5 (41.6%) Level 4–1 (8.3%) Level 5–6 (50%) | |
| Brighton level 1–3 | 11 (91.7%) | 6 (50%) | *p* = 0.03 |
| Medical intervention | 8 (66.7%) | 6 (50%) | ns |
| Episode of anaphylaxis in the past | 10 (83.3%) | 9 (75%) | ns |
| Diagnostics to LA *n* = 9 positive | 1 (8.3%) | 3 (25%) | ns |

AE, angioedema; LA, local anesthetics; NSAIDs, nonsteroidal anti-inflammatory drugs; PAR, perennial allergic rhinitis; PF, paraformaldehyde; SAR, seasonal allergic rhinitis.

## 4. Discussion

According to actual recommendations, the use of paraformaldehyde as a dental material should be significantly limited; however, in certain cases, paraformaldehyde paste is still used. Its use can cause delayed anaphylaxis, which can be life-threatening. Paraformaldehyde is a precursor of formaldehyde. It releases formaldehyde gas, which may be used as a disinfectant [8]. Formaldehyde released through dentine has a destructive effect on periodontal and bone tissues [14,15]. Formaldehyde is also a hapten most commonly causing a type IV reaction. However, anaphylactic type I reactions were also reported after use of this substance [11,12]. Analysis of our material showed that anaphylaxis is not incidental cases and should always be considered during dental treatment.

In the present study, we analyzed a group that consisted of 157 patients. All of the patients experienced hypersensitivity reaction during dental treatment. None of the dentists who treated these patients suspected hypersensitivity to paraformaldehyde. In almost all cases, they recommended further allergological workup into the direction of local anesthetic hypersensitivity or other substances used in dentistry, but not to paraformaldehyde. In 152 patients, local anesthesia was applied during the treatment. Clinical history revealed that in 24 patients, paraformaldehyde was used, and in 12 patients, we found the presence of a specific IgE against paraformaldehyde. Moreover, 2 of the 12 patients showed a positive patch test to formaldehyde. In this group, we conducted a detailed clinical analysis of anaphylactic reaction symptoms. One patient experienced an anaphylactic reaction twice after contact with paraformaldehyde. The first type IV reaction occurred during a visit to the hairdresser, and the second type I reaction occurred after dental treatment. It was reported that types I and IV allergies coexist in individual cases [12].

A group of 12 patients showing no specific IgE against paraformaldehyde constituted the control group. The case and control groups were not statistically different in terms of the prevalence of atopic diseases and drug hypersensitivity. The studied groups statistically differed in terms of the time of occurrence of the anaphylactic reaction. In the case group, hypersensitivity reactions most frequently occurred more than 1 h after paraformaldehyde application. In the control group, reactions were most frequently noted up to 1 h after treatment. We reviewed the literature describing cases with anaphylaxis/angioedema caused by formaldehyde in root canal disinfectants and found that about 1/2 of the reported cases developed symptoms over 2 to 24 h after dental treatment. We speculated that the delay in the manifestation of her symptoms was possibly due to gradual formaldehyde release from paraformaldehyde and time of penetrating and diffusing of formaldehyde

outside the dentin [16]. It was shown that formaldehyde outside the tooth is detectable 30 min after treatment [17].

Almost all patients in the study group did not have any previous history of formaldehyde allergy. Prolonged exposure to gas formaldehyde may cause IgE-mediated immune response. The gas may be released from formaldehyde-containing products, such as pressed wood materials produced with urea–formaldehyde resins, commonly found indoors [18]. According to Gu Y et al., it may be possible that formaldehyde gas indirectly induces long-term IgE-mediated allergic responses but through the mechanism of NK-cell activation [19].

Erythema and pruritus were significantly more frequent in the study group. The incidence of urticaria, angioedema, dyspnea, throat obstruction, weakness, and RR drop did not vary between groups. The likelihood of anaphylactic reactions according to the Brighton scale was significantly higher in the study group. This suggests that paraformaldehyde-induced type I reaction can be life-threatening. None of the patients treated with paraformaldehyde was informed of the possibility of an anaphylactic reaction of delayed onset.

The present research should be evaluated in the context of its limitations. One of them is the retrospective design of the study. However, prospective analysis was not suitable in this case because, according to the International Consensus on Drug Allergy, screening subjects without a prior history of allergic drug reactions is not recommended [20].

Another limitation is the relatively small number of patients who were allergic to paraformaldehyde and covered in our analysis. Considering these limitations, the present study should be regarded as a pilot one. The obtained results encourage us to continue the research in this field.

Moreover, our analysis was focused on the adult patient population because, in our hospital, only adult patients were diagnosed. It is worth mentioning that late-onset anaphylaxis due to paraformaldehyde may also affect pediatric patients. Paraformaldehyde as a dental material is a component of formocresol used in pediatric dentistry for pulpotomy [21].

## 5. Conclusions

Patients who received paraformaldehyde during dental treatment should be informed of the possibility of an anaphylactic reaction, which might occur even 24 h after treatment. Attention to the risk of anaphylactic reaction after paraformaldehyde use may protect the patient from a repeat reaction during subsequent treatment.

**Author Contributions:** Conceptualization, E.J.-S., R.G., and J.G.; methodology, J.G.; validation and formal analysis, J.G.; investigation, L.R.; resources, L.R.; data curation, L.R. and J.G.; writing—original draft preparation, E.J.-S., L.R., and J.G.; writing—review and editing, E.J.-S., L.R., J.G., and R.G.; visualization, J.G.; supervision, J.G.; project administration, J.G. All authors agreed to be accountable for all aspects of the work in ensuring that questions related to the accuracy or integrity of any part of the work are appropriately investigated and resolved. All authors have read and agreed to the published version of the manuscript.

**Funding:** This research received no external funding.

**Institutional Review Board Statement:** Not applicable.

**Informed Consent Statement:** Not applicable.

**Data Availability Statement:** Not applicable.

**Conflicts of Interest:** The authors declare no conflict of interest.

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
