# Peer review of "Immediate Hypersensitivity Reactions to Paraformaldehyde Used as a Dental Material"

_coatings, doi:10.3390/coatings12101493_

Round 1

Reviewer 1 Report

Very relevant information. Congrats!

Please add a comment, in the discussion section, on the formaldehyde component of formocresol, that is used in pediatric dentistry.

Author Response

Dear Sir/Madam,

We appreciate the time and effort that You have dedicated to providing valuable feedback on our manuscript titled: “Immediate hypersensitivity reactions to paraformaldehyde used as a dental material.” We are grateful for Your insightful comment on our paper. Please find revised version of manuscript with highlighted corrections suggested by You and other reviewers.

We look forward to hearing from You regarding our submission and to respond to any further questions and comments You may have.

Sincerely Yours,

Lesia Rozłucka on behalf of all the co-authors

Reviewer 2 Report

The authors of the manuscript “Immediate hypersensitivity reactions to paraformaldehyde used as a dental material” made an error by submitting this manuscript to the journal Coatings. The manuscript didn’t fit the journal aims and quality.

The aim of the study described in manuscript was to assess the frequency of immediate-type hypersensitivity and to analyze the course of the reaction after the use of dental materials containing paraformaldehyde. The target journal must be one dedicated to the dentistry, being a “single-center retrospective observational case-control study included patients…”.

No details about materials and coatings appear in the manuscript.

The release of formaldehyde is a widely known effect that occurs with many materials used in dentistry, but the authors didn’t mention some typical examples. 

Before resubmission to another journal, I suggest to authors to make some improvements by adding some important and useful information like sources of formaldehyde, exposure limits, current methods to assess formaldehyde release from dental materials including direct instrumental analysis and derivatization methods involving subsequent chromatography and ultraviolet detection, as well as more sensitive methods using fluorescence.

Author Response

Dear Sir/Madam,

We appreciate the time and effort that You have dedicated to providing valuable feedback on our manuscript titled: “Immediate hypersensitivity reactions to paraformaldehyde used as a dental material.” We are grateful for Your insightful comments on our paper. Prepared manuscript has been submitted to the special issue of “Coatings” titled “Advances and Innovations in Dental Materials and Coatings”. Authors focused on clinical aspects of hypersensitivity to paraformaldehyde as one of the materials commonly used in dentistry, rather than its technical aspects. However, after Your important suggestions we added information about typical examples of dental materials including paraformaldehyde. Please find revised version of manuscript with highlighted corrections suggested by You and other reviewers. We look forward to hearing from You regarding our submission and to respond to any further questions and comments You may have.

Sincerely Yours,

Lesia Rozłucka on behalf of all the co-authors

Reviewer 3 Report

COMMENTS ARE PROVIDED IN THE ATTACHED DOCUMENT. KINDLY IMPLEMENT THE SAME

Author Response

Dear Sir/Madam,

We appreciate the time and effort that You have dedicated to providing valuable feedback on our manuscript titled: “Immediate hypersensitivity reactions to paraformaldehyde used as a dental material.” We are grateful for Your insightful comments on our paper. We have been able to incorporate changes to reflect most of the suggestions. Please find revised version of manuscript with highlighted corrections suggested by You and other reviewers.

We look forward to hearing from You regarding our submission and to respond to any further questions and comments You may have.

Sincerely Yours,

Lesia Rozłucka on behalf of all the co-authors.

Round 2

Reviewer 2 Report

The manuscript could be published if the special issue editors consider relevant for their topic.

Reviewer 3 Report

Dear Author,

My comments have been addressed well. I recommend the revised draft be accepted